# EFFICIENT AND EFFECTIVE UNCERTAINTY QUANTIFICATION FOR LLMS

**Miao Xiong**[*]
National University of Singapore

**Andrea Santilli**[*]
Sapienza University of Rome

**Michael Kirchhof**[*]
University of Tübingen

**Adam Goliński**
Apple

**Sinead Williamson**
Apple

## ABSTRACT

Uncertainty quantification (UQ) is crucial for ensuring the safe deployment of large language model, particularly in high-stakes applications where hallucinations can be harmful. However, existing UQ methods often demand substantial computational resources, e.g., multi-sample methods such as Semantic Entropy (Kuhn et al., 2023) usually require 5-10 inference calls, and probing-based methods require additional datasets for training. This raises a key question: *How can we balance UQ performance with computational efficiency?* In this work, we first analyze the performance and efficiency of various UQ methods across 6 datasets $\times$ 6 models $\times$ 2 prompt strategies. Our findings reveal that: 1) Multi-sample methods generally perform only marginally better than single-sample methods, i.e., $\leq 0.02$ in AUROC over 65% settings, despite significantly higher inference costs. 2) Probing-based methods perform well primarily on mathematical reasoning and truthfulness benchmarks, while multi-sample methods only show a clear advantage on knowledge-seeking tasks. These findings suggest that the high computational cost does not translate into significant performance gains. Despite their similar overall performance, we observe only moderate correlations between different UQ methods, suggesting they may be capturing different uncertainty signals. This motivates us to explore the potential of combining different methods to harness their complementary strengths at lower computational costs. Our experiments demonstrate that a simple combination of single-sample features can match or even outperform the existing best-performing methods. These findings suggest a promising direction for developing cost-effective uncertainty estimators.

## 1 INTRODUCTION

Large Language Models (LLMs) have achieved remarkable advancements in recent years, with applications spanning various high-stakes domains such as healthcare and law (Hadi et al., 2024; Lambert et al., 2024; Zhao et al., 2024). However, an increasing body of research has identified LLMs' tendency to hallucinate, generating nonfactual or misleading content (Ji et al., 2022; Huang et al., 2023). Ensuring the safe deployment of LLMs in these critical areas requires effective mitigation of such hallucinations. A key approach for hallucination detection is to quantify the uncertainty in LLM outputs (Tian et al., 2023; Xiong et al., 2024). By accurately quantifying the uncertainty associated with each model output and flagging high-uncertainty responses for expert review (Geifman and El-Yaniv, 2017), the reliability of LLM-based systems can be substantially improved.

While uncertainty quantification (UQ) is crucial for enhancing LLM trustworthiness, balancing its effectiveness with computational efficiency remains a challenge. Existing UQ methods for LLMs can be broadly categorized into three types based on their computational cost: single-sample, multi-sample, and probing-based methods. *Single-sample* methods, such as perplexity (Jelinek et al., 1977), estimate uncertainty based on the next token prediction distributions with minimal computational overhead, as token probabilities are byproducts of the LLM's generation process. In contrast,

---

[*]Work done during an internship at Apple.

*multi-sample* methods, which require multiple inference calls (e.g., $M = 10$) to capture the diversity within possible answer spaces, impose a much higher computational cost. Some multi-sample methods also have additional costs, e.g., Semantic Entropy (Farquhar et al., 2024) requires clustering answers based on their semantic meaning. Finally, *probing-based* methods (Liu et al., 2024; Kadavath et al., 2022) require a separate dataset to train an additional classifier for predicting the correctness of an answer.

Recent work has held multi-sample methods up as a gold standard of UQ (Kuhn et al., 2023; Farquhar et al., 2024). Indeed, methods such as Semantic Entropy, which approximates the model's "belief distribution" by sampling multiple predictions, offer a much more solid justification for use as an uncertainty method than single-sample methods such as perplexity. However, a practical question to ask is: *Is the increased cost of these methods justified by the improved performance?* To explore the trade-off between performance and method complexity, we evaluate the performance of a variety of single-sample, multi-sample and probing methods across 6 datasets covering commonly used tasks such as mathematical reasoning, knowledge-seeking, and truthfulness benchmarks, and models ranging from 7B to 70B parameters across both 0-shot and 5-shot settings (see §2).

Our experiments reveal several key insights: (1) While multi-sample methods outperform basic single-sample methods, the performance improvement is often modest, i.e., $\leq 0.02$ in AUROC for most cases, despite consuming significantly more inference calls ($10\times$ in our experiments). (2) Probing-based methods consistently outperform both single-sample and multi-sample approaches in mathematical and truthful benchmarks. (3) Multi-sample methods tend to perform better on knowledge-seeking datasets, whereas for math and truthful benchmarks, single-sample and probing-based methods have comparable or better performance. These findings suggest that the high computational cost does not always translate into consistent and significant performance gains.

Diving deeper into the relation between single-sample, multi-sample, and probing-based methods, we observe low to moderate correlations between the ten uncertainty methods, suggesting different methods might capture different notions of uncertainty. This hypothesis raises the question: *can we combine single-sample metrics to leverage their strengths with low computational cost*? We study the problem through the lens of a supervised classifier. Specifically, we aggregate available features from various methods and train a classifier on a small held-out dataset to predict the correctness of model outputs. The classifier's prediction score is then treated as a new uncertainty quantification score, allowing us to evaluate the impact of combining these methods (see §5). In this way, we find that: 1) Simply combining single-sample features consistently matches or outperforms the best-performing existing methods, suggesting a promising direction for cost-effective uncertainty estimators. 2) Incorporating all available features from single-sample, probing-based, and multi-sample methods results in the highest performance, consistently surpassing existing methods. These findings underline the potential for developing more resource-efficient uncertainty quantification methods by fully utilizing the information from existing methods.

## 2 LLM UNCERTAINTY QUANTIFICATION METHODS AND EVALUATIONS

### 2.1 LLM UNCERTAINTY METHODS

The goal of uncertainty quantification methods for LLMs is to provide a scalar value indicating how likely a given answer is to be incorrect. Uncertainty quantification methods for LLMs can be broadly categorized into three types from the computational cost perspective: single-sample, multi-sample, and probing-based methods. We choose multiple representatives from each of the types here and outline all the methods we have considered in this paper below.

**Notation** Let $x = \{x_1, ..x_N\}$ denote the sequence of generated tokens, and $x_{\text{prompt}}$ denote the sequence of tokens corresponding to the input prompt; $\hat{p}(\cdot)$ denote the probability assigned by the model to a token or a sequence of tokens.

**Single-sample methods** We consider single-sample methods to be methods that a) do not require multiple samples from the model, and b) do not require additional training. As such, the computational cost is negligible compared to the LLM inference process.

*Perplexity*   is the exponentiated mean of the token log likelihood of the output sequence $x$:

$$\exp\left\{-\tfrac{1}{N}\sum_{i=1}^{N}\log\hat{p}(x_i|x_{<i},x_{\text{prompt}})\right\}.$$

When ignoring the exponential and just caring about the order of the final values, as AUROC does, this is equivalent to a length-penalized version of sequence probability.

*Negative Sequence probability* (Seq. Prob.)   computes the (negative) joint token probability of the generated output sequence $x$:

$$\hat{p}(x|x_{\text{prompt}}) = \prod_{i=1}^{N}\hat{p}(x_i|x_{<i},x_{\text{prompt}}).$$

*Mean token entropy* (M.T. Entropy, Fomicheva et al., 2020)   is the average per-token entropy of the output sequence, which can be seen as a single-sample estimate of the predictive entropy (see Eq. 9 of Malinin and Gales, 2020):

$$\tfrac{1}{N}\sum_{i=1}^{N}\mathcal{H}(\hat{p}(x_i|x_{<i},x_{\text{prompt}})).$$

**Multi-Sample Methods**  We consider multi-sample estimators to be those that require multiple samples $x^{(s)}, s = 1,\ldots, M$, from the model and do not require additional training. Therefore, they have high inference cost ($\geq M C_{\text{infer}}$ where $C_{\text{infer}}$ is the cost for one inference call).

*Regular Entropy* (Reg. Ent, sometimes Naive Entropy, Farquhar et al., 2024)   computes the entropy over $M$ sampled responses:

$$-\sum_{s=1}^{M}\hat{p}(x^{(s)}|x_{\text{prompt}})\log\hat{p}(x^{(s)}|x_{\text{prompt}})$$

where $x^{(s)}$ denote the $s$-th sample out of $M$ sampled responses, and $\hat{p}(x^{(s)}|x_{\text{prompt}})$ is the sequence probability for the sampled response $x^{(s)}$.

*Semantic Entropy* (S.E., Melamed, 1997; Farquhar et al., 2024)   clusters sampled outputs via bidirectional entailment into semantic concepts $c_1,\ldots,c_C$, and estimates each concept's probability $\hat{p}(c_i)$ as the sum of the sequence probabilities inside that cluster. Following (Farquhar et al., 2024), we use Deberta-Large-MNLI (He et al., 2020) as our entailment model, and the cluster probabilities are then used to estimate the entropy of the underlying semantic concept distribution:

$$SE(x) = -\sum_{i=1}^{C}\hat{p}(c_i|x_{\text{prompt}})\log\hat{p}(c_i|x_{\text{prompt}}).$$

*Discrete semantic entropy* (Discrete S.E.)   is an alternative estimator for semantic entropy proposed by Farquhar et al. (2024), which uses the relative class frequencies as an estimate for $p(c_i)$, without accounting for the sequence probabilities.

*Number of clusters* (Num. Clusters)   is a heuristic estimator of uncertainty, introduced by Lin et al. (2023). It looks at the number of clusters generated by the entailment method used for Semantic Entropy, and has the same inference cost.

*P(True)* (Kadavath et al., 2022)   solicits the LLM to assess truthfulness of a statement. It first generates multiple samples from the LLM, then construct a prompt presenting these as possible answers (see Appendix B), before asking the LLM to decide whether the greedily decoded answer is true or false, recording the probability of the response "True".

**Probing-based Methods**  Probing-based methods usually train a lightweight model on the model's embeddings or its entire architecture to predict the answer's correctness. As such, they require an additional training dataset and some additional cost at training time: $C_{\text{train, probe}}$, where $C_{\text{train, probe}}$ is the cost of training the probe and $C_{\text{train, probe}} \ll C_{\text{train, LLM}}$. However, they do not introduce extra inference cost. The most commonly used method is P(I know), referred to as *P(IK) Probe* in this paper. The original implementation (Kadavath et al., 2022) fine-tunes the entire network. We follow Kapoor et al. (2024) and train a logistic regression probe on the last layer embedding representation of the final token.

## 2.2   EVALUATION SETTINGS

To evaluate the performance of various uncertainty quantification methods across a diverse set of application cases, we consider several key variables: datasets, models, and prompt types.

**Datasets** We evaluate the performance of UQ methods across three types of tasks. **Mathematical reasoning** datasets comprise human-written math word problems. We use the GSM8K (Cobbe et al., 2021) and SVAMP (Patel et al., 2021) datasets. **Knowledge-seeking** datasets are "trivia"-style questions about knowledge factoids. We use three datasets spanning different domains, TriviaQA (Joshi et al., 2017), Natural Questions (NQ, Kwiatkowski et al., 2019), and PopQA (Mallen et al., 2022). **Truthfulness** tasks, represented by TruthfulQA (Lin et al., 2021), targets common misconceptions that people might hold to test whether LLMs are affected as well.

**Models** To ensure that our findings on uncertainty quantification methods are broadly applicable, we cover different model families (e.g., Llama, Falcon), parameter size (e.g., 7b, 40b, 70b) and training objectives (e.g., base vs. instruct): **Llama-2-7b-chat**, **Meta-Llama-3-8B-Instruct**, **Meta-Llama-3-8B**, **Llama-2-13b-chat**, **Llama-2-70b-chat**, **falcon-40b-instruct**.

**Prompt Types** Following Farquhar et al. (2024), we look at both 0-shot and 5-shot settings, with examples selected at random from the training set. See Appendix B for the specific prompt.

**Evaluation Metrics** We evaluate the UQ performance using Area Under the Receiver Operating Characteristic curve (AUROC, Mucsányi et al., 2024), which measures how well the uncertainty methods distinguish between correct and incorrect answers. Specifically, if correct answers tend to have lower uncertainty and incorrect answers higher uncertainty, the AUROC will be high, otherwise, it will be low. For reference, random guessing yields an AUROC of 0.5. In our results, we report the average AUROC across the settings outlined above. To compute AUROC, we follow Farquhar et al. (2024) to generate answers with $T = 0.1$ and use `Meta-Llama-3-8B-Instruct` as the grader to determine correctness, and then calculating AUROC accordingly.

## 3  DO MORE EXPENSIVE ESTIMATORS GIVE BETTER UNCERTAINTIES?

Recent works have held multi-sample methods up as a gold standard for uncertainty quantification (Lakshminarayanan et al., 2017; Kuhn et al., 2023; Farquhar et al., 2024). Indeed, methods such as Semantic Entropy, which approximates the posterior distribution over latent concepts by sampling multiple predictions, offer a more solid justification for use as an uncertainty method than those logit-based single-sample methods such as perplexity. However, a practical question to ask is: *Is the increased cost of these methods justified by the improved performance?*

To explore the trade-off between performance and cost, we compare the performance between a variety of single-sample, multi-sample, and probing methods across the 6 datasets × 6 models × 2 prompts = 72 experiment settings. In Figure 1a, for each setting, we plot the best AUROC of the single-sample methods against the best AUROC of the multi-sample methods, with the marker shape indicating the identity of the best overall method. In Figure 1b, we repeat this visualization, this time including the probing-based method alongside the single-sample methods. Based on the experiment results, we make the following findings:

**1. While multi-sample methods outperform single-sample methods, the increase in performance is not dramatic.** As shown in Figure 1a, while on average the best multi-sample method does outperform the best single-sample method, i.e., many points lie above the diagonal, there are many scenarios where the single-sample methods perform better. Overall, both multi-sample and single-sample methods cluster fairly closely around the diagonal, with many settings falling within a ±0.02 gap range (the grey area, corresponding to the average standard error obtained via bootstrap sampling), highlighting that the best-performing single-sample approaches achieve comparable results to the best-performing multi-sample methods across most datasets and configurations. This is especially notable given the multi-sample methods' $M \times$ higher computational cost needed to generate $M$ responses (in our case, $M = 10$) for every question, which does not translate into dramatic performance gains in AUROC. This phenomenon is further highlighted in Figure 5, where the average AUROC of single-sample methods remain quite close to that of multi-sample methods in both knowledge-seeking and math reasoning settings, and in the truthful setting, the best single-sample methods even outperform the multi-sample methods.

**2. Probing-based methods consistently outperform both single-sample and multi-sample methods on mathematical and truthfulness benchmarks.** Across math-related and truthfulness datasets in Figure 1b, colored in orange, red, and green, the probing-based P(I Know), represented by star-shaped markers, achieves higher AUROC scores than both single-sample and multi-sample

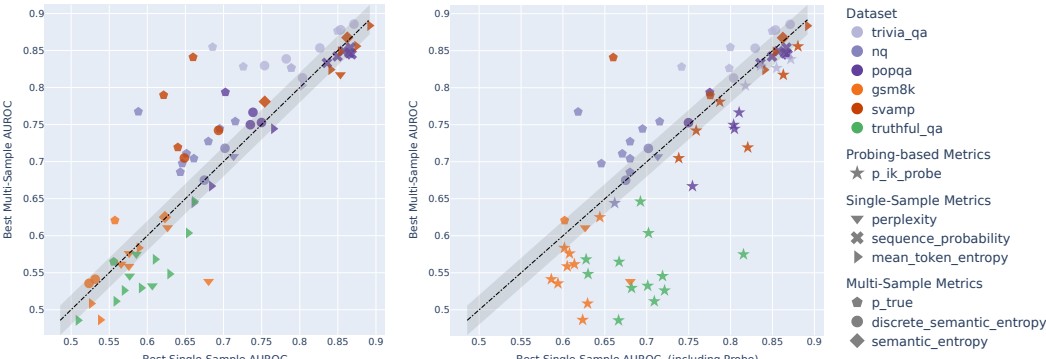

(a) Probing-based methods not included.    (b) Probing-based methods included.

Figure 1: Performance comparison of single-sample and multi-sample methods (with and without probing-based methods included alongside single-sample methods). Each dot represents the best AUROC that single-sample methods (x-axis) and multi-sample methods (y-axis) can achieved under a specific dataset, model, and prompt setting. Among them, the color indicates the dataset, the shape denotes the best-performing method for that configuration. Points above the $x = y$ line indicate settings where the best-performing method was a multi-sample method; points below the $x = y$ line indicate settings were the best-performing method was a single-sample (or probing-based in Figure 1b)) method. Notably, many dots cluster within the $y = x \pm 0.02$ dashed line, which corresponds to the average standard error (obtained using bootstrapped samples). Additionally, the prevalence of star-shaped markers in the right plot highlights the strong performance of the probing-based method, especially in truthful and math reasoning datasets.

methods. This suggests that probes are more effective at capturing the uncertainty in reasoning-based questions and questions where the answer does not directly correspond to the model's parametric memory. For TruthfulQA, where P(I Know) dramatically outperforms the other methods, we hypothesize that the advantage stems from the inherent structure of this dataset. Ground truth answers often follow clear patterns such as "there is no scientific evidence for ..." and "Nothing would happen if ...". This consistent structure is likely captured when training the probe on the dataset.

**3. Multi-sample methods show the clearest advantage on knowledge-seeking datasets.** Contrarily, the configurations with AUROC scores above the diagonal, where multi-sample methods perform better, are mostly from knowledge-seeking datasets (TriviaQA, NQ, and PopQA). This indicates that multi-sample methods tend to provide more meaningful improvements in performance for knowledge-seeking tasks.

## 4 EXPLORING THE RELATIONSHIP BETWEEN DIFFERENT UQ METHODS

One interesting observation from Section 3 is that, despite their different mechanisms and varying degrees of theoretical justification, the different UQ methods exhibit notably similar performance across a variety of tasks. This suggests two possible hypotheses: 1) These methods inherently capture the **same** underlying uncertainty signal from the model, albeit in different forms, resulting in similar AUROC scores; or 2) These methods capture **orthogonal** uncertainty signals, but coincidentally have similar overall performance. To explore which hypothesis is more likely, we measure the Spearman's rank correlation of the uncertainty scores produced by the nine different UQ methods. Figure 2 shows the average rank correlations across all 72 experimental settings.

In Figure 2, we observe that the Semantic Entropy estimator and its variants, including the Semantic Entropy-based heuristic of Number of Clusters, are highly correlated, suggesting they capture similar information. Similarly, perplexity and mean token entropy (M. T. Entropy) are highly correlated, suggesting that they are dominated by the effect of a few important tokens. They are also fairly highly correlated with regular entropy, which makes sense since both mean token entropy and regular entropy can be seen as estimators of the overall distributional entropy. However, we see fairly low correlations between P(True), P(I Know), and the other methods. Overall, we find that there are distinct clusters, but that the inter-cluster correlations are only moderate. We find that the correla-

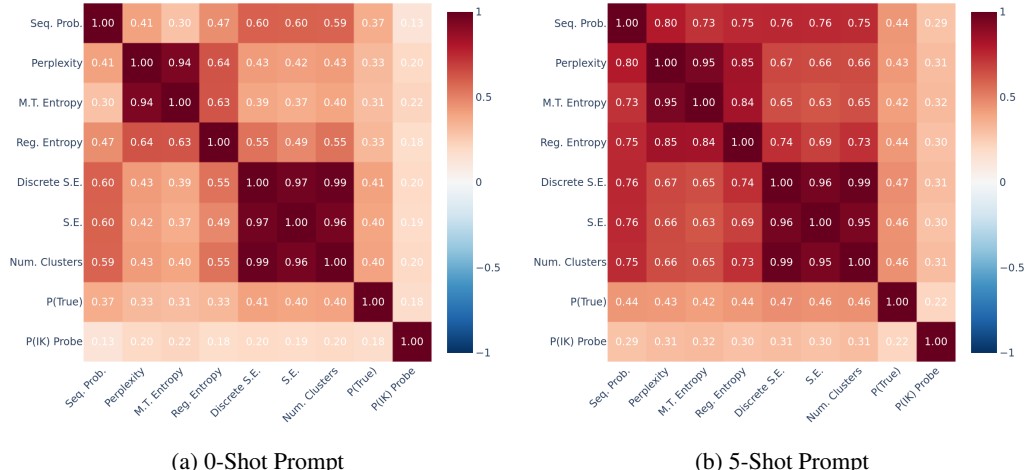

(a) 0-Shot Prompt              (b) 5-Shot Prompt

Figure 2: Spearman's correlation between ten UQ methods for the 0-shot prompt strategy (**Left**) and the 5-shot prompt strategy (**Right**). Each square displays the average Spearman's correlation between each pair of UQ methods across 72 settings. In general, the correlations between many pairs of methods are low to moderate (i.e., below 0.6), particularly in the 0-shot settings, indicating that different methods might capture various aspects of uncertainty. Please see Figure 6 for more details including the standard deviation.

tion between methods is higher when we provide 5-shot examples with short answers (Figure 2b) than in the zero-shot setting, likely in part due to less variation in response length; however even in this setting we see a clear difference between inter-cluster and intra-cluster correlation.

The fact that we do not see high correlation between all metrics, despite seeing similar performance across metrics, offers some support for the second hypothesis, namely that the different uncertainty estimators indeed capture different uncertainty signals.

## 5 COMBINING DIFFERENT UQ METHODS TO LEVERAGE THEIR STRENGTHS

In Section 4, we found that the correlations between many pairs of methods were only moderate, despite these methods performing comparably in practice. This raises the possibility that they are, in part, capturing complementary information about the model's uncertainty. If true, this would allow us to combine different UQ methods to obtain more accurate uncertainty estimation. To evaluate this hypothesis, we aggregate available features from various methods, and use these features to train a linear classifier on the held-out data to predict the correctness of model outputs, and then use the classifier's predictive probability as the new uncertainty score. This enables us to explore the best potential performance we can gain from integrating the information from various UQ methods.

We are particularly interested in exploring *low-cost* uncertainty estimators that can provide state-of-the-art uncertainty quantification. To this end, we group our features into four categories, based on the cost of obtaining the features:

- **Single-sample methods (SINGLE).** In addition to perplexity, sequence probability, and mean token entropy, we include summaries of the token probability distribution output by the model, inspired by Gupta et al. (2024). Specifically, given an output sequence $S = \{x_0, x_1, ..., x_N\}$ with $N$ tokens, we extract the token probability $\hat{p}(x_i|x_{\text{prompt}})$ and token entropy $H(\hat{p}(x_i|x_{\text{prompt}}))$ for each token $x_i$. We then compute the 0%, 25%, 50%, 75%, and 100% quantiles for the sequences of token probabilities and token entropies, resulting in vectors $(p_0, p_{25}, p_{50}, p_{75}, p_{100})$ and $(\mathcal{H}_0, \mathcal{H}_{25}, \mathcal{H}_{50}, \mathcal{H}_{75}, \mathcal{H}_{100})$, respectively. We also include the sequence length $N$.

- **Probing-based methods (PROBE).** Here, we consider P(I Know). This implicitly includes information about the model embeddings.

- **Non-cluster-based multi-sample methods (MULTI).** Here, we consider those multi-sample methods that do not involve clustering the output. We treat these separately from the cluster-based methods since they are cheaper at inference time, since the bi-directional entailment adds an additional $2M(M-1)$ LLM calls in order to cluster $M$ samples. Here, we include regular entropy and P(True). We also look at distributional information across the samples: for the $M$ sampled responses, we obtain sequence probability and perplexity for each of the responses, sort them into 2 vectors, and include them in our features.

- **Cluster-based multi-sample methods (CLUSTER).** Here, we include the three methods that require semantic entailment: Semantic Entropy, Discrete Semantic Entropy, and the number of clusters. We also include 4 additional methods based on Semantic Entropy: 1) Maximal cluster probability, i.e., the sum of sequence probabilities of all samples in this largest cluster, 2) Maximal cluster ratio, i.e., the portion of samples in the largest cluster relative to all clusters, 3) Greedy cluster probability, and 4) Greedy cluster ratio. Note that the difference between 1 & 3 and 2 & 4 is that the first two use the largest cluster while the latter two use the cluster containing the greedy response we are evaluating.

To evaluate the contribution of different type of features, we propose 5 types of combination: SINGLE-only, SINGLE+PROBE, SINGLE+MULTI, SINGLE+MULTI+CLUSTER and all features (ALL). The SINGLE-only combination assesses whether different token probability information can be combined in a complementary manner to obtain a cheap, high-quality estimator. The remaining estimators look at which pairs of features (e.g., model embeddings, multi-sample clusters) can lead to the best performance. We train a logistic regression classifier from `scikit-learn` (Pedregosa et al., 2011) with a maximum of 1000 iterations. We apply greedy feature selections to select a maximum of 10 features. To reduce the additional overhead of the held-out dataset, we limit the use of validation set samples for training the combination algorithm to be maximal 1000 samples (500 for training the classifier and 500 as a validation set to select the features). Note that the above 1000-sample validation set is randomly sampled from the training data used for P(I Know) and is disjoint from the test data. Therefore, compared to probing-based methods, we do not introduce any new training samples but only use a subset of their training data.

**Findings** The average AUROC over all configurations for each uncertainty method and our proposed metric combination are shown in Figure 3 and Figure 4.

**1. Simply combining single-sample features (SINGLE-only) consistently matches or outperforms the best-performing existing methods, suggesting a promising direction for cost-effective uncertainty estimators.** For knowledge-seeking datasets (Figure 3a, b, c), the average AUROC of SINGLE-only is comparable to (TriviaQA, NQ) or outperforms (PopQA) the best-performing multi-sample methods, represented by the purple bars. This finding is more obvious through the lens of the lowest average rank in Figure 4. For math reasoning datasets (Figure 3d,e), SINGLE-only also shows comparable performance to probe-based methods (represented by the orange bar). These findings are promising since the SINGLE-only logistic regression has lower-dimensional inputs than P(I Know) and does not require extracting the latent representations, making it an option for grey-box models.

**2. Metric combination adapts well to different datasets, overcoming limitations of multi-sample and probing-based methods in certain contexts.** While the performance of P(I Know) is notably strong in datasets like TruthfulQA and SVAMP, it underperforms in knowledge-seeking datasets such as NQ. However, our SINGLE-only metric combination does not suffer from these limitations, consistently providing strong performance across both knowledge-seeking and other types of datasets. In cases where the probing method excels (e.g., TruthfulQA and SVAMP), the combination of SINGLE+PROBE features achieves even better results using the same amount of training data.

**3. Combining single-sample, probe, and multi-sample methods results in robust performance, consistently surpassing existing methods.** By incorporating single-sample-based, probing-based, and multi-sample features (represented by SINGLE+PROBE, SINGLE+MULTI+CLUSTER, ALL), we observe improved performance across most datasets. In datasets such as GSM8K and TruthfulQA, this combination yields results that either surpass or are comparable to the best-performing existing methods. This demonstrates that integrating multiple uncertainty estimation methods can maintain or improve performance relative to existing methods, across diverse datasets and settings.

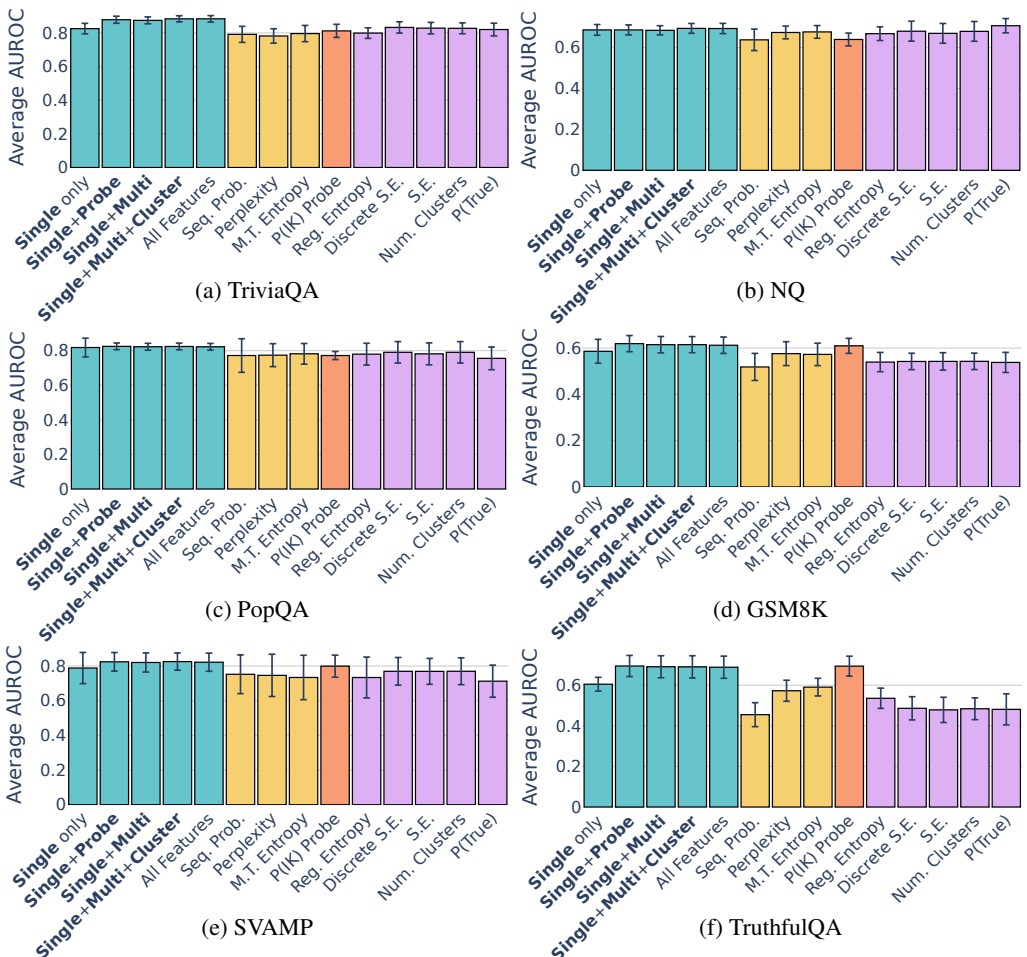

Figure 3: Average AUROC comparison of five feature combination methods and existing ten UQ methods across various settings. Each bar represents the average AUROC for a specific method across all task-related datasets, models, and prompt settings, with error bars showing standard deviation. The orange bars represent single-sample methods, the teal bar represents the probing-based method, and the purple bars represent multi-sample methods. SINGLE-only consistently matches or outperforms the best-performing existing methods, suggesting a promising direction for cost-effective uncertainty estimators. Please see Figure 4 for the average rank plot.

## 6 CONCLUSION

Accurate uncertainty quantification with minimal computational cost is essential for the safe and practical deployment of LLMs in real-world applications. In this paper, we analyzed three types of UQ methods: single-sample, multi-sample, and probing-based. Surprisingly, despite the substantial computational overhead—often requiring 5-10 times more inference calls or additional training datasets—we did not observe significant performance improvements. In most cases, single-sample methods perform comparably to multi-sample methods and even to probing-based methods on knowledge-seeking datasets. This highlights the need to fully leverage existing computational resources and develop more cost-effective approaches. In our exploration of this direction, we find that fully utilizing all available information from a single inference shows the potential to outperform or match the best-performing existing methods. This work is just a first step in this direction; we hope future work will explore further the utility of metric combination methods outside of the narrow scope of question answering. One limitation of our approach is that training a task-specific classifier limits generalizability of our approach; an interesting avenue for future direction is the development of training-free algorithms that combine multiple uncertainty systems without the risk of overfitting.

ACKNOWLEDGMENTS

We are deeply grateful to researchers in the team for their invaluable discussions. We also want to thank Arno Blaas and Eugene Ndiaye for their helpful feedback on the paper.

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

## A  RELATED WORKS

**Uncertainty Quantification Methods.**    Uncertainty quantification methods for LLMs can be broadly categorized into three types from the computational cost perspective: single-sample, multi-sample, and probing-based methods. *Single-sample* methods, usually requiring one or two inference calls, have two types: logit-based and verbalized. Logit-based, such as perplexity (Jelinek et al., 1977), sequence probability (Fomicheva et al., 2020), and mean token entropy (Fomicheva et al., 2020) mainly estimate uncertainty based on the next token prediction distributions. The computational cost is negligible, as token probabilities are a byproduct of the LLM's generation process. While verbalized methods (Lin et al., 2022; Xiong et al., 2024; Tian et al., 2023) rely on model's self-evaluation by prompting LLMs to explicitly express their its uncertainty in the output during the same inference call or another separate inference call. In contrast, *multi-sample* methods require multiple inference calls (e.g., $M = 10$) to capture the diversity of possible answer spaces, leading to computational costs scaling by at least a factor of $M$ compared to single-sample methods. In some cases, obtaining uncertainty estimates from the $M$ samples incurs additional computational cost; for example, calculating Semantic Entropy (Farquhar et al., 2024) and its variants requires us to first cluster answers based on their semantic meaning. *Probing-based* methods Liu et al. (2024); Kadavath et al. (2022) extract internal model representations and train probes to predict the correctness of an answer. However, this approach requires a held-out dataset for probe training, adding further computational overhead.

**Uncertainty Quantification Benchmarks**   There are multiple benchmarking papers for uncertainty quantification methods for LLMs. Among them, Xiong et al. (2024); Tian et al. (2023) primarily examines black-box models with a focus on verbalized uncertainty, while Ye et al. (2024) emphasizes performance variations across different models and datasets. In contrast, our work provides a different perspective by focusing on the computational cost and the correlation between different methods, aiming to explore whether these methods capture distinct notions of uncertainty. There is a concurrent work Valentin et al. (2024) working on cost-effective uncertainty quantification methods by giving a limited computational budget and solving the optimization problem to find the set of features to use.

## B  PROMPT TEMPLATES

Here we include all the prompts we have used in this work.

**P(True)**   The P(True) method we use in this paper is the 5-shot self-evaluation method.

---

**Prompt for P(True)**

Question: {question}
Brainstormed Answers:
{list of regenerated answers}
Possible answer: {original answer}
Is the possible answer:
A) True
B) False
The possible answer is: {' A' if the answer is correct else ' B'}
*Repeat the above templates using 5 randomly selected training samples.*
Question: {question}
Brainstormed Answers:
{list of regenerated answers}
Possible answer: {original answer}
Is the possible answer:
A) True
B) False
The possible answer is:

---

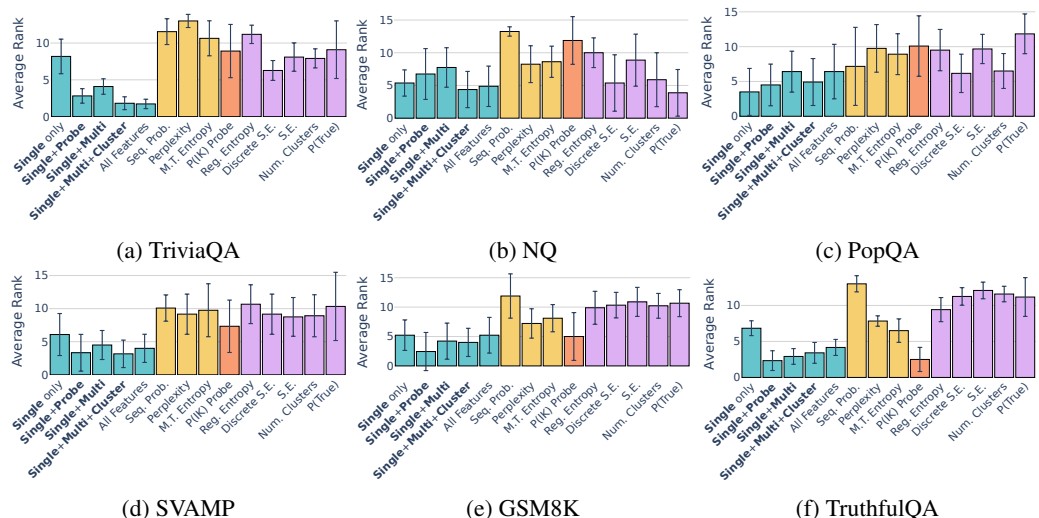

|  (a) TriviaQA | (b) NQ | (c) PopQA |
| --- | --- | --- |
| (d) SVAMP | (e) GSM8K | (f) TruthfulQA |

Figure 4: Average rank comparison of five feature combination methods and existing ten UQ methods across various settings. Each bar represents the average rank for a specific method across all task-related datasets, models, and prompt settings, with error bars showing standard deviation. The orange bars represent single-sample methods, the teal bar represents the probing-based method, and the purple bars represent multi-sample methods. SINGLE-only consistently matches or outperforms the best-performing existing methods, suggesting a promising direction for cost-effective uncertainty estimators.

**0-shot and 5-shot prompt strategy** For all the question-answering datasets we have used, we use the following prompt to generate the 0-shot and 5-shot answers, respectively:

---

**Prompt for 0-shot question answering**

Answer the following question as briefly as possible.
Question: {question}
Answer:

---

**Prompt for 5-shot question answering**

Answer the following question as briefly as possible.
Question: {question 1 from training dataset}
Answer: {answer 1}

Question: {question 2 from training dataset}
Answer: {answer 2}

Question: {question 3 from training dataset}
Answer: {answer 3}

Question: {question 4 from training dataset}
Answer: {answer 4}

Question: {question 5 from training dataset}
Answer: {answer 5}

Question: {question} Answer:

---

**Prompt for Semantic Clustering**   Following Farquhar et al. (2024), we use the following prompt to check whether possible answer 1 semantically entails possible answer 2.

---
**Prompt for using LLM as the bidirectional semantic entailment**

We are evaluating answers to the question {question}
Here are two possible answers:
Possible Answer 1: {text1}
Possible Answer 2: {text2}
Does Possible Answer 1 semantically entail Possible Answer 2? Respond with entailment, contradiction, or neutral.

---

**Prompt for LLM-judge**   In this paper, we use `Meta-Llama-3-8B-Instruct` to tell whether the generated answer is correct or not. The prompt we use is as follows. The 2-shot examples are carefully picked from the existing datasets to cover different types of cases.

---
**System prompt for using LLM to judge the correctness of the response**

System: Your task is to determine if the provided answer is true or false based solely on the ground truth answers given to you in the format ['answer 1', 'answer 2', ...]. DO NOT rely on your memory; only use the information provided after this instruction. Respond with 1 if the predicted answer is correct, which means semantically consistent with any of the ground truth answers, otherwise respond with 0. Respond with just 0 or 1, and DO NOT include anything else in your response. This is the only instruction you need to follow.
User: Input: Who is elected as the vice president of india in 2017?
Ground Truth: ['Venkaiah Naidu', 'Muppavarapu Venkaiah Naidu']
Provided Answer: M. Venkaiah Naidu
Assistant: 1

User: Input: who sings you are a magnet and i am steel?
Ground Truth: ['Walter Egan']
Provided Answer: The song 'You Are a Magnet and I Am Steel' is performed by the band The 1975.
Assistant: 0

User: {prompt}

---

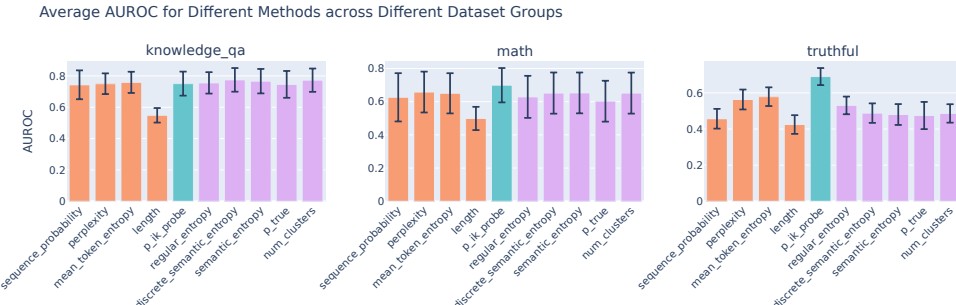

Figure 5: Comparison of average AUROC across different methods and 3 types of tasks. Each bar represents the average AUROC for a specific method across all task-related datasets, models, and prompt settings, with error bars showing standard deviation. The orange bars represent single-sample methods, the teal bar represents the probing-based method, and the purple bars represent multi-sample methods. Notably, the probing-based method outperforms others in math reasoning datasets and TruthfulQA, especially in TruthfulQA, where it consistently leads. For knowledge-seeking datasets, performance between single-sample and multi-sample methods is similar, with minor AUROC differences.

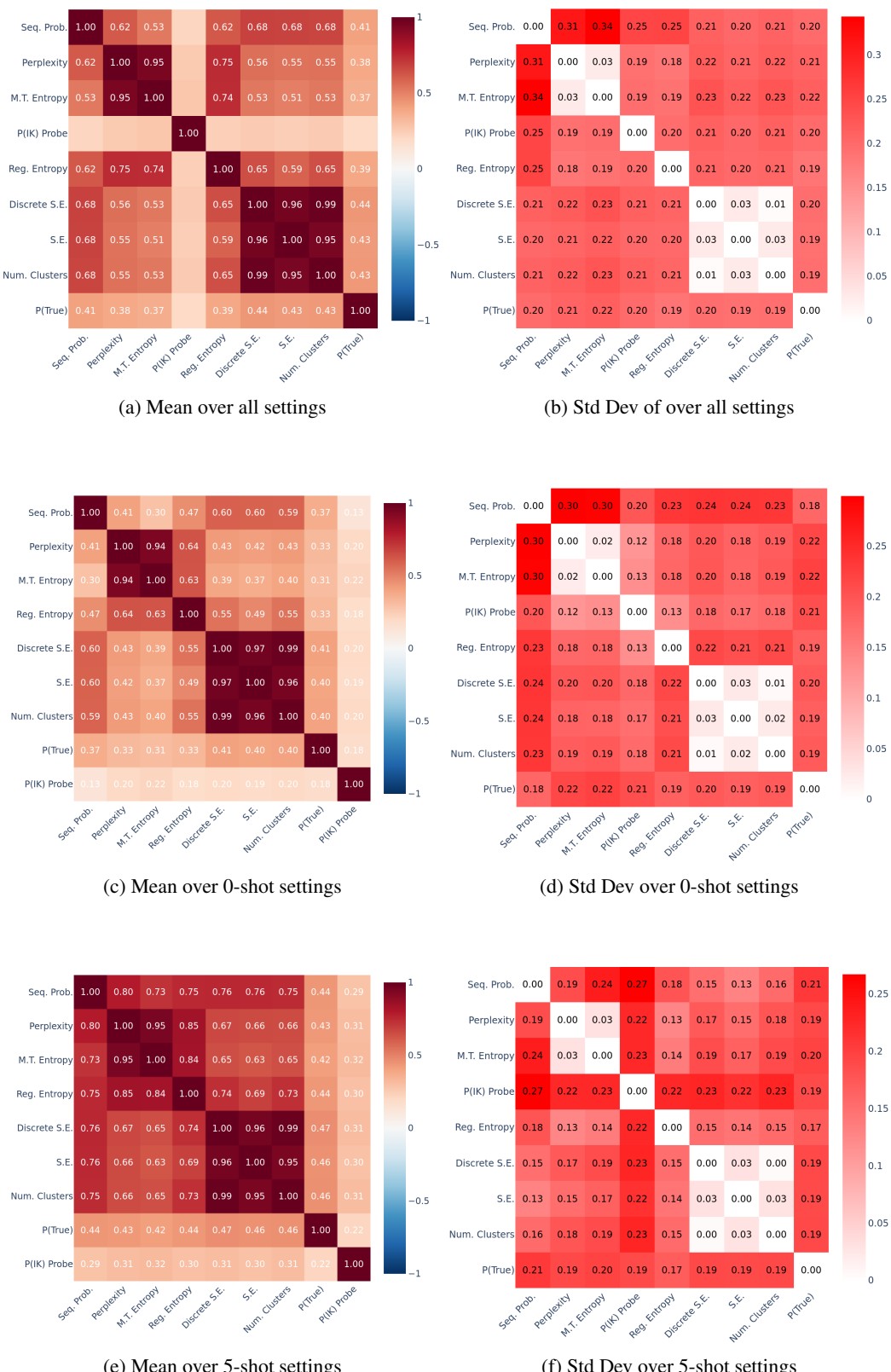

Figure 6: The average mean (**Left**) and standard deviation (**Right**) of Spearman's correlation between ten UQ methods for 72 configurations (**First Row**), the 0-shot prompt strategy (**Second Row**) and the 5-shot prompt strategy (**Third Row**). Each square displays the average Spearman's correlation between each pair of UQ methods across 72 settings, with correlations above 0.35 annotated with text. In general, the correlations between many pairs of methods are low to moderate (i.e., below 0.6), particularly in the 0-shot settings, indicating that different methods might capture different aspects of uncertainty.

