# OpenReview forum: "Efficient and Effective Uncertainty Quantification for LLMs"
_NeurIPS.cc/2024/Workshop/SafeGenAi — SafeGenAi Poster_

### Official Review · Reviewer_saKA · 2024-10-09
**Review of "Efficient and Effective Uncertainty Quantification for LLMs"**

**Rating:** 5
**Confidence:** 3

**Review:**

### **Summary**

The paper proposes an efficient and effective approach for uncertainty quantification (UQ) in large language models (LLMs), particularly addressing the high computational costs associated with existing UQ methods. It explores single-sample, multi-sample, and probing-based methods across various datasets, finding that while multi-sample methods only slightly outperform single-sample methods, they do so at a much higher computational cost. The paper introduces a method that combines single-sample features to achieve similar or better performance than multi-sample approaches, offering a more cost-effective solution.

---
### **Strengths**

1. The paper provides a comprehensive evaluation of different UQ methods across multiple models, datasets, and prompt settings, offering insights into the performance and cost trade-offs.
2. By focusing on single-sample methods and their combinations, the paper presents a practical approach that significantly reduces computational expenses without compromising performance.
3. The idea of combining single-sample, probing-based, and multi-sample methods into a unified framework to leverage complementary strengths is simple but shows good performance.

---
### **Weaknesses**

1. While the integration approach is promising, the individual techniques explored (single-sample, multi-sample, and probing-based) are already well-established. The paper’s primary contribution appears to be the combination rather than the development of new methods.

2. The paper emphasizes computational cost throughout, but it does not include experiments analyzing or quantifying the computational complexity. The experiments focus solely on performance, which is insufficient for an analytical study that aims to address efficiency concerns.

3. The paper claims that probing-based methods require additional datasets and impose computational overhead. However, in practice, the training of probes can be extremely fast, depending on the size of the probe itself. Therefore, this aspect may not be as significant a limitation as suggested.

---

### Official Review · Reviewer_cED6 · 2024-10-09
**The paper shows that single-sample UQ methods can match multi-sample ones in efficiency and performance, with promising insights for real-world use**

**Rating:** 7
**Confidence:** 5

**Review:**

**Pros** :
- well written
- clear explanation of experiments and settings
- clear interpretation of results
- Comprehensive Analysis
- novel insights

**Cons**:
- Fig. 1 is hard to understand (it would be better to make it clearer)
- Limited Discussion on Limitations

---

### Official Review · Reviewer_HMHU · 2024-10-09
**nice paper, novel analysis**

**Rating:** 9
**Confidence:** 3

**Review:**

The paper compares existing LLM uncertainty quantification methods regarding their computational cost and performance. The paper then compare the difference between signals learned by different methods and test combined methods according to the correlation of signals. The paper is well written. The analysis of computational cost is novel, but the significance of combined method is not convincing enough.

Pros
1. The paper is clearly written and presented.
2. It is novel to compare the tradeoff between computational cost and performance of uncertainty quantification methods.

Cons
1. Section 4 is not clear. What signal is compared in the correlation plots? Distribution of predicted uncertainty?
2. The improvement of combined methods in Fig 3 does not seem significant.